# Which Infectious Diseases Drive the Highest Absenteeism Costs—An Analysis Based on National Data Covering the Entire Polish Population in the Period of 2018–2023

**DOI:** 10.3390/healthcare13182284

**Published:** 2025-09-12

**Authors:** Michał Seweryn, Grzegorz Juszczyk, Marcin Czech

**Affiliations:** 1Faculty of Medicine, Andrzej Frycz Modrzewski Krakow University, 30-705 Krakow, Poland; 2Department of Public Health, Medical University of Warsaw, 02-091 Warsaw, Poland; grzegorz.juszczyk@wum.edu.pl; 3Pharmacoeconomic Department, Institute of Mother and Child, 01-211 Warsaw, Poland; marcin.czech@pw.edu.pl

**Keywords:** infectious diseases, absenteeism, indirect costs, sickness absence, productivity loss, COVID-19, economic burden

## Abstract

**Background:** Infectious diseases pose a serious epidemiological and economic challenge for all healthcare systems. However, there is a lack of comprehensive analyses assessing the cost of absenteeism attributable to all infectious diseases. Our objective was to evaluate the burden of absenteeism-related costs due to infectious diseases in comparison with other major public health challenges. **Methods:** We applied the human capital approach to estimate the indirect costs of absenteeism caused by infectious diseases in Poland between 2018 and 2023. In particular, we assessed the relative contribution of different groups of infectious diseases to the overall economic burden. Data were obtained from the Social Insurance Institution (ZUS). **Results:** The total cost of absenteeism due to infectious diseases in Poland during the six-year period was EUR 5.3 billion. Over 78% of these costs were attributed to pneumonia and other acute lower respiratory tract infections (ICD-10: J12–J22): EUR 1.89 billion, COVID-19 (ICD-10: U07–U09): EUR 1.82 billion, and influenza (ICD-10: J09–J11): EUR 444.5 million. Infectious diseases imposed a greater economic burden in terms of absenteeism than each of the three conditions used as comparators: malignant neoplasms, depression, and ischemic heart disease. **Conclusions:** Our six-year analysis of sickness absence in Poland indicates that infectious diseases—particularly during the COVID-19 pandemic—are major drivers of productivity loss. When compared with other leading public health challenges, their economic burden is substantial. These findings underscore the importance of investing in preventive measures, particularly vaccination programs.

## 1. Introduction

Although significant advances have been made in the prevention and management of infectious diseases, their global health burden remains substantial. While the incidence of some conditions has declined over time, especially in well-developed countries, new infectious threats continue to emerge. Recently, the COVID-19 pandemic has clearly demonstrated that our control over infectious diseases is limited and can be rapidly challenged by novel pathogens [1]. The pandemic period also contributed to an erosion of public trust in vaccination in many countries, including Poland, thereby exacerbating challenges in maintaining effective epidemiological control [2].

Infectious diseases exert a multifaceted impact on society. Their often-rapid progression poses a direct threat to the health and lives of affected individuals, while their ease of transmission and potential for uncontrolled spread represent a serious public health risk.

Beyond clinical outcomes, infectious diseases generate a substantial economic burden, consisting of both direct medical costs (hospitalization, pharmacotherapy, outpatient care) and indirect costs related to productivity loss through absenteeism, presenteeism, informal caregiving, and premature mortality. Cost-of-illness methodology emphasizes that a full societal perspective should capture these categories [3,4]. In the working-age population, indirect costs are often dominant: for example, systematic reviews have shown that the majority of influenza-related costs in adults aged 18–64 years arise from productivity losses [5,6]. Moreover, evidence from respiratory outbreaks demonstrates considerable burdens, not only for healthcare systems, but also for employers and households [7].

In addition to their clinical consequences, infectious diseases impose a substantial economic burden, generating both direct costs for the healthcare system and indirect costs associated with productivity losses—most notably through sickness absence.

To the best of our knowledge, there is no published study that comprehensively examines the indirect costs of sickness absence due to all infectious diseases being considered as a single group. The existing literature tends to focus on individual diseases or selected clusters, such as influenza [8,9], COVID-19 [10,11], or pneumococcal infections [12].

However, by full economic burden, we refer to the total costs borne by society, including direct medical costs, direct non-medical costs (transport, informal care), indirect costs (absenteeism, presenteeism, premature mortality, permanent or temporary work disability), and intangible costs measured as lost quality and length of life (QALY/DALY) [3,4,13]. Absenteeism is therefore only one element of the burden: guidelines and systematic reviews recommend also capturing presenteeism, time spent by informal caregivers, and losses due to premature death [14,15]. For example, estimates for the United States show that seasonal influenza generates costs of about USD 11.2 billion annually, of which ~USD 8 billion are due to indirect costs [16]. Similarly, a Polish analysis demonstrated that in 2016 influenza led to approximately PLN 7.1 million in direct hospitalization costs but as much as PLN 161.6 million in productivity losses, confirming that indirect costs largely outweigh direct medical expenditures [17].

This highlights a gap in understanding the full economic burden of infectious diseases from a broader societal perspective, particularly in Central and Eastern Europe, where such analyses remain limited.

To further clarify what we mean by “social and economic costs”, we emphasize that burden-of-disease studies show that infections such as influenza, tuberculosis, or pneumococcal disease remain among the leading causes of lost DALYs in Europe [18]. At the same time, Global Burden of Disease estimates indicate that in Poland in 2021, communicable diseases accounted for nearly 15% of all DALYs, with the majority represented by Years of Life Lost (YLLs) rather than Years Lived with Disability (YLDs) [19]. This highlights that, beyond the short- and medium-term economic consequences of absenteeism, infectious diseases continue to impose a substantial long-term health burden through premature mortality. In addition, reviews emphasize the disruptive effect of airborne and droplet-borne infections on enterprises and entire sectors, where high absenteeism and reduced productivity contribute significantly to economic loss [20]. For example, recent EU-level assessments suggest that long COVID alone reduced labor supply by ~0.2–0.5% in 2021–2022 [21].

Understanding the social and economic costs associated with infectious diseases is essential for evidence-based public health planning. Among various types of indirect costs, productivity losses due to absenteeism represent one of the most tangible and measurable consequences of infectious morbidity. Workforce health has increasingly gained recognition as a critical component of the public health and economic research agenda. Organizations are becoming more aware that maintaining employees’ health not only reduces healthcare expenditures but also contributes to sustained productivity and competitiveness. As a result, there has been a growing emphasis on identifying and managing productivity losses linked to employee health issues [22]. One of the most prominent forms of such losses is absenteeism, defined as the failure of employees to attend scheduled work due to illness or other health-related reasons. Assessing the magnitude of these losses can help inform healthcare policy, shape preventive strategies, and support broader efforts to improve population health resilience.

In this study, we aim to address this gap by quantifying the indirect costs of infectious disease-related sickness absence in Poland between 2018 and 2023, using national-level administrative data from the Social Insurance Institution (ZUS). Specifically, we focus on assessing the relative contribution of various infectious disease groups to the total economic burden of absenteeism. Additionally, we compare these costs with those associated with other major public health challenges in Poland, including cancer depression and ischemic heart disease.

The analysis is conducted from a societal perspective, capturing productivity losses reflected in reductions in gross domestic product (GDP). By highlighting the scale of absenteeism caused by infectious diseases, we aim to foster dialog among stakeholders and contribute to discussions on the need to strengthen prevention efforts, particularly immunization programs. We also seek to advance the scientific discourse by emphasizing that infectious diseases should be regarded not only as epidemiological and clinical challenges, but also as a substantial economic burden with wide-ranging societal impacts.

## 2. Materials and Methods

In our study, we used the most recent data covering the period from 2018 to 2023. This extended time horizon enables us to analyze the situation both before and after the COVID-19 pandemic, which is particularly valuable in light of the study’s objectives. 

The human capital approach (HCA) was adopted in this study to estimate the indirect costs associated with sickness absence, as it remains the most commonly used method in health economic analyses across Europe, particularly in countries with less flexible labor markets and strong public insurance systems. It is especially relevant in studies relying on administrative data, such as those from national social insurance institutions [23]. In addition, the choice of HCA reflects the recommendations of the Polish HTA guidelines [24], which indicate this method as the standard for cost-of-illness and economic evaluations. In our analysis, we primarily relied on sickness absence data from the Social Insurance Institution (ZUS) [25], complemented by GDP, workforce size, and related statistics from the Central Statistical Office (GUS) [26].

In this study, sickness absence was defined strictly as medical leave certified by a physician and recorded in the ZUS system with an ICD-10 diagnostic code. The dataset therefore reflects only the personal illness of the insured employee. Other categories of absence, such as caregiving allowances or administrative quarantine/isolation orders during the COVID-19 pandemic, are recorded by ZUS under separate schemes and were not included in the analysis. 

All cost values presented in euros (EUR) were converted from Polish zloty (PLN) using average annual exchange rates published by the National Bank of Poland (NBP). Year-specific exchange rates were applied to preserve temporal accuracy and to reflect actual market conditions during each year. All parameters used in the model are presented in Table 1: Overview of socioeconomic parameters used in the model to estimate productivity losses due to infectious diseases in Poland from 2018 to 2023.

The absenteeism cost $\left(C_{abs}\right)$ was estimated using an adjusted GDP-based human capital approach, in which the GDP per employee was scaled by a labor productivity correction factor (β = 0.65) to reflect the marginal contribution of labor to economic output. This value is consistent with methodological standards commonly applied in Poland [24] and corresponds to the labor share parameter used in the Cobb–Douglas production function. It is also employed in macroeconomic models developed by the European Commission, including those featured in the Ageing Report [27].

We used the following formula to calculate the cost of absenteeism $\left(C_{abs}\right)$:$$C_{abs}=\left(\frac{{GDP}_{year}}{N_{empl}}\times \beta \right)\times \left(\frac{1}{N_{workdays}}\right)\times D_{missed}$$

${GDP}_{year}$ = Gross domestic product in the selected year (EUR);$N_{empl}$ = Number of employed individuals in the selected year;β = Correction factor (0, 65);$N_{workdays}$ = Number of working days in the selected year;$D_{missed}$ = Number of workdays missed due to disease (aggregated).

Infectious diseases were identified using ICD-10 codes A00–B99. Additionally, selected codes outside this range were included to capture diseases of infectious origin with high absenteeism impact, such as influenza (J09–J11) and COVID-19 (U07.1–U07.2). A detailed list of the included ICD-10 codes is provided in the Results Section.

To contextualize the burden of sickness absence due to infectious diseases, we compared them with selected non-communicable conditions representing major public health concerns in Poland. Specifically, we included malignant neoplasms (ICD-10 codes C00–D09), ischemic heart disease (ICD-10 codes I20–I25), and depression (ICD-10 codes F33–F34). These diagnoses were chosen because they are leading contributors to morbidity and mortality in Poland and are also associated with substantial sickness absence. For each of these conditions, we applied the same methodological approach as for infectious diseases, relying on administrative data from the Social Insurance Institution (ZUS) to calculate sickness absence days, complemented by macroeconomic indicators from the Central Statistical Office (GUS) to estimate productivity losses. This ensured consistency and comparability across disease categories. 

For comparative purposes, we supplemented our analysis of sickness absence with Global Burden of Disease (GBD 2021) estimates of DALYs, YLLs, and YLDs for selected major disease groups. This allowed us to situate the indirect costs of infectious diseases within the broader spectrum of population health losses.

## 3. Results

### 3.1. Sickness Absence Due to Infectious Diseases

The number of sick leave days due to infectious diseases in Poland varied between 2018 and 2020 (Table 2) and was significantly influenced by the impact of the COVID-19 pandemic. Sick leave days increased from over 7.5 million in 2018 and more than 6.1 million in 2019 to 10.2 million in 2020—a rise of 65% compared to 2019 and 35% compared to 2018.

In 2018, over 75% of all sick leave days were due to pneumonia and other acute lower respiratory tract infections (J12–J22), which accounted for 58.9%, and influenza (J09–J11), which contributed 16.24%. In 2019, these conditions were responsible for over 70% of sick leave days—59.29% for J12–J22—while the proportion due to influenza decreased to 11.05%. The year 2020 brought significant changes. The leading cause of sickness absence became COVID-19 (U07–U09), which accounted for 47.32% of all sick leave days. This pushed J12–J22 into second place, with a share of 26.05%. Another clear effect of the pandemic was a marked increase in sickness absence due to conditions in the B25–B34 group, which in 2020 represented 7.41% of all sick leave days. Within this group, the ICD-10 code B34—”Viral infection of unspecified site”—was particularly notable, responsible for nearly 96% of these cases. The number of B34 cases was 14 times higher than in 2019 and 18 times higher than in 2018. Influenza accounted for just 6.5% of sick leave days in the first year of the pandemic.

Between 2021 and 2023, the total number of sick leave days remained high, reaching 9.8 million in 2021 and nearly 10.4 million in 2022, before a noticeable decrease was observed in 2023 (Table 3). In 2021, COVID-19 accounted for more than half of all sick leave days (54.97%), while the proportion of sick leave days attributed to pneumonia and other acute lower respiratory tract infections (ICD-10 codes J12–J22) remained similar to that of 2020, at 26.78%. The unexpectedly favorable epidemiological situation in Poland in 2021, characterized by a low incidence of influenza cases, led to an almost fivefold reduction in sick leave days due to influenza (J09–J11), which comprised only 1.44% of all sick leave days. Although the restrictions, obligations, and prohibitions related to the state of epidemic were lifted in Poland in May 2022 [28], sick leave days related to diagnoses associated with the virus (ICD-10 codes U07–U09) remained a significant issue that year, accounting for the highest share of total sick leave days at 47.53%—a level comparable to that observed during the first year of the pandemic (2020). Sick leave days attributed to pneumonia and other acute lower respiratory tract infections (J12–J22) remained stable at 26.54%. The number of sick leave days due to influenza (J09–J11) increased more than fourfold, accounting for 6.01% of all absenteeism related to infectious diseases. The year 2023 marked the first time that sick leave days due to COVID-19 (29.7% of all sick leave days related to infectious diseases) were no longer the leading cause, having been overtaken by diagnoses classified under J12–J22, which accounted for 34.66%. A notable rise in influenza cases (J09–J11) also contributed to a significant increase in related sick leave days, representing 12.5% of the total. However, these three causes have remained dominant, accounting for the majority of sick leave days due to infectious diseases between 2021 and 2023—83.19% in 2021, 80.08% in 2022, and 76.87% in 2023.

### 3.2. Productivity Losses from Infectious Diseases

Productivity losses resulting from infectious diseases have been increasing in Poland over the past six years, largely driven by the global health crisis caused by the COVID-19 pandemic. The total value of absenteeism-related costs ranged from EUR 567.3 million in 2019 to EUR 1.15 billion in 2022 (Table 4). 

Over the six-year observation period, nearly 70% of sickness absence costs, above EUR 3.7 billion, were attributable to pneumonia and other acute lower respiratory tract infections (J12–J22), which accounted for 35.63% of the total, and COVID-19 (U07–U09), which contributed 34.3%. Influenza (J09–J11) ranked third, with total costs amounting to EUR 444.54 million, representing 8.38% of the overall burden.

Intestinal infectious diseases (A00–A09) ranked fourth in terms of absenteeism-related costs, generating EUR 363.96 million and accounting for 6.86% of the total. Within this group, over 90% of the costs were attributable to A08: Viral and other specified intestinal infections (EUR 107.14 million) and A09: Diarrhea and gastroenteritis of presumed infectious origin (EUR 230.79 million). 

The B00–B09 category, which includes viral infections with skin and mucous membrane lesions, also represented a considerable share of absenteeism costs during the study period—exceeding 5% of the total, with losses reaching above EUR 270 million. Within this group, sickness absences related to shingles (B02) and chickenpox (B01) were particularly prominent, together accounting for nearly 82% of the category’s total costs. Over the six-year period, the financial burden of shingles amounted to EUR 162.37 million, while chickenpox-related losses reached EUR 58.5 million.

The B25–B34 group, classified as other viral infections, accounted for 4.68% of total absenteeism costs, generating EUR 248.6 million in losses. A marked increase in this group was observed during the pandemic, and notably, this elevated share persisted into 2023. This trend may suggest that certain cases of COVID-19 or influenza were coded within this broader category. This hypothesis is supported by the dominance of diagnoses B33: other viral diseases, not elsewhere classified, and B34: viral infection of unspecified site, which together accounted for EUR 234.45 million, representing nearly 95% of the group’s total costs.

The remaining disease categories contributed substantially less to the overall burden. Tuberculosis (A15–A19) accounted for 1.94% of total costs, and other bacterial diseases (A35–A49) for 1.45%. HIV and viral hepatitis combined represented less than 1% of absenteeism-related costs due to infectious diseases. Similarly, negligible contributions were observed for the following groups: infections predominantly transmitted through sexual contact (A50–A64): 0.1%; viral infections of the central nervous system (A80–A89): 0.4%; and bacterial and viral agents as causes of diseases (B95–B97): 0.18%.

Regardless of the impact the SARS-CoV-2 pandemic may have had on these cost data, it is important to emphasize that many of the health conditions contributing significantly to absenteeism costs are diseases for which effective primary prevention measures—namely, vaccinations—are available. This includes, among others, influenza, COVID-19, respiratory syncytial virus (RSV), and infections caused by pathogens such as *Streptococcus pneumoniae* (pneumococcus), *Haemophilus influenzae*, and *Neisseria meningitidis* (meningococcus), as well as the varicella zoster virus, which causes herpes zoster (shingles).

### 3.3. Infectious Diseases as a Public Health Challenge: Comparison with Other Major Conditions

To illustrate the broader public health burden associated with sickness absence due to infectious diseases, we compared their impact with that of selected non-communicable diseases that represent major health priorities in Poland. Specifically, we focused on malignant neoplasms (ICD-10 C00:D09), ischemic heart disease (ICD-10 I20:I25), and depression (ICD-10 F33:F34)—conditions that, while differing in etiology, are associated with substantial levels of sickness absence and productivity loss among affected individuals.

In addition, to contextualize the magnitude of these costs within the wider framework of population health, we present burden-of-disease estimates expressed as disability-adjusted life years (DALYs). According to the Global Burden of Disease (GBD) 2021 study, communicable diseases still accounted for nearly 15% of all DALYs in Poland, with the majority attributable to Years of Life Lost (YLLs) rather than Years Lived with Disability (YLDs). By contrast, non-communicable diseases such as cancer and ischemic heart disease remain the leading causes of DALYs overall. Table 5 summarizes these figures, highlighting the relative contributions of infectious and non-infectious conditions to the total burden of disease in Poland.

These comparisons illustrate that, although the short- and medium-term economic burden of infectious diseases is largely driven by sickness absence and productivity loss, their overall health impact remains substantial when measured in DALYs. This dual perspective underscores the importance of integrating both economic and epidemiological indicators when designing prevention strategies and setting public health priorities.

Depression is increasingly acknowledged in Poland, as in other industrialized countries, as a critical public health concern, closely linked to diminished work performance, including reduced self-assessed work capacity, sickness absence, and occupational burnout [29]. It represents a significant yet often under-recognized independent risk factor for work-related disability, affecting up to 20% of individuals over their lifetime and ranking among the leading causes of disability and impaired quality of life [30]. As shown in Figure 1, although depression is associated with a substantial number of sickness absence days, infectious diseases generate higher overall costs. This trend is evident both during the COVID-19 pandemic and in non-pandemic years; however, the cost difference was markedly smaller in the years preceding the pandemic.

In Poland, cancer poses an escalating burden on the healthcare system, with profound social and economic implications. It ranks as the second leading cause of death and remains the foremost contributor to premature mortality (deaths before the age of 65), especially among women [31]. The economic burden of cancer in Poland is significantly amplified by indirect costs resulting from productivity losses due to sickness absence, long-term disability, and premature mortality. In 2009, these losses were estimated to exceed 0.8% of the national GDP [32]. Our analysis indicates that sickness absence costs in Poland due to infectious diseases are substantially higher than those associated with malignant neoplasms (Figure 2). Over the six-year period, these costs varied considerably, with social losses during the pandemic due to sickness absence from infectious diseases reaching up to twice the level observed for cancer-related sickness absences.

The Global Burden of Disease Study 2021 (GBD 2021) reports that ischemic heart disease (IHD) continues to be the foremost cause of mortality and morbidity globally, accounting for 8.99 million deaths in 2021 [33]. Despite numerous public health efforts over the past decades, IHD remains a major global health challenge [34]. Previous analyses indicate that permanent and temporary work incapacity is the primary cost driver of cardiovascular diseases in Poland, with indirect costs exceeding direct costs by more than fivefold during the same period [35]. Our analysis (Figure 3) demonstrates that although ischemic heart disease represents a significant health concern, the costs of sickness absence due to infectious diseases in Poland between 2018 and 2023 were more than two to over four times higher.

A comparison of sickness absence costs reveals that infectious diseases impose a substantially greater economic burden than other major public health issues. Neither cancer, ischemic heart disease, nor depression—which affects a large segment of the population and contributes significantly to sickness absence—generate costs comparable to those caused by communicable diseases. The COVID-19 pandemic further underscored the profound impact that epidemiological crises can have on these costs.

While comparing infectious diseases with other major public health challenges provides useful context, important limitations must be acknowledged. Depression, for example, is known to be underdiagnosed and stigmatized, leading to underreporting and consequently lower levels of registered sickness absence [36,37]. Malignant neoplasms, on the other hand, frequently result in permanent disability, early retirement, or exit from the labor market, rather than repeated episodes of short-term sickness absence [38]. Similarly, ischemic heart disease may cause both acute sickness absence and long-term reductions in productivity due to disability or presenteeism [39]. Therefore, comparisons restricted to short-term sickness absence likely underestimate the full indirect costs associated with these conditions, including long-term disability, informal care, and productivity impairment beyond recorded absences. These differences should be borne in mind when interpreting the relative burden across disease groups.

## 4. Discussion

Our findings align with a growing body of evidence indicating that infectious diseases—particularly respiratory illnesses such as COVID-19, influenza, and pneumonia—are significant contributors to productivity losses across diverse healthcare systems. Similar studies conducted in Europe and worldwide highlight the extent of this burden and reinforce the critical importance of effective preventive measures, especially vaccination and workplace health policies. 

For example, Fisman et al. [8] highlighted that influenza and influenza-like illnesses lead to substantial productivity losses in both Europe and North America, with estimated indirect costs per case often exceeding direct medical expenditures. These findings are consistent with our observations on influenza-related sickness absence in Poland, where indirect costs exceeded EUR 444.5 million between 2018 and 2023 (Table 4). 

A retrospective study from Greece by Lampropoulos et al. [11] quantified COVID-19-related sickness absence using the human capital approach and reported substantial economic losses, particularly during the peak of the pandemic. They stressed that such indirect costs can rival or exceed healthcare spending in high-transmission scenarios [11]. This reinforces our results, wherein COVID-19 alone accounted for over EUR 1.8 billion in sickness absence costs—surpassing even cancer and ischemic heart disease. 

Furthermore, Golicki et al. explored the indirect costs of pneumococcal disease in Poland using similar ZUS data and found that sickness absence accounted for a considerable share of the overall economic burden [12]. Their results highlight the broader value of vaccination strategies targeting *Streptococcus pneumoniae*, which also features prominently in our dataset under J12–J22. Similarly, the significant costs related to shingles and varicella infections (B00–B09), which together exceeded EUR 270 million, relate to earlier studies emphasizing the cost-effectiveness of varicella zoster virus vaccination. 

Beyond infectious diseases, comparative analyses with non-communicable diseases (NCDs) such as cancer and depression show that the economic burden of communicable diseases—especially during epidemic periods—can be even more pronounced. The Global Burden of Disease (GBD) 2021 study corroborates that indirect costs from infectious disease morbidity are often underestimated, particularly when presenteeism and long-term disability are excluded [33]. Importantly, Schmid et al. (2017) demonstrated that organizational leadership and workplace culture significantly influence the prevalence and cost of both absenteeism and presenteeism, suggesting that beyond epidemiological measures, systemic changes in workplace policy could further mitigate the economic impact of infections [22]. 

However, a systematic review by Webster et al. [40] reported that the prevalence of infectious illness-related absenteeism ranges from 35% to 97%, with higher rates observed among healthcare and social care workers. The reasons for this behavior are multifactorial and include organizational factors (e.g., lack of paid sickness absence, workplace culture), job-related characteristics (e.g., workload, staffing shortages), and personal motivations (e.g., fear of job loss, sense of professional obligation). Further research by Daniels et al. [41] found that the prevalence of work presenteeism ranged from 14.1% to 55% for confirmed respiratory infections, and from 6.6% to 100% among individuals with suspected or subclinical infections. Given the wide variability in prevalence across occupational settings, further research is needed to better understand its impact on indirect costs. 

When analyzing healthcare costs, direct costs are the most intuitive and widely recognized category. Indirect costs, on the other hand, are more abstract, as they do not represent actual financial outlays but rather reflect potential losses to society caused by a patient’s reduced ability to participate in economic activities [42]. Nevertheless, it is important to raise awareness that indirect costs also constitute societal expenses and should be carefully considered, particularly in healthcare decision-making. 

Furthermore, a potential limitation of our analysis is the use of the human capital approach (HCA) to estimate productivity losses. While the HCA is consistent with European standards and the Polish HTA guidelines, and facilitates international comparability, it may overestimate costs compared to the friction cost method (FCM). However, in our study this risk is mitigated by the use of conservative assumptions, as we did not include presenteeism, informal caregiving, or intangible costs. Therefore, our estimates should be viewed as a cautious rather than inflated reflection of the economic burden of infectious diseases.

Nevertheless, limitations in our study must be recognized. One of the most important is the reliance on administrative data, which may not fully capture disease burden due to underreporting, diagnostic misclassification, or varied sickness absence behavior. As highlighted by Mikos et al. (2020), refusal to take sick leave despite infection remains a real concern in Poland, potentially distorting the true economic picture [43]. 

Additionally, our estimates rely on a labor productivity correction factor of 0.65, consistent with previous Polish and European cost-of-illness studies. While this choice is well-founded, we acknowledge that alternative assumptions (e.g., 0.6–0.7) could affect the absolute magnitude of costs. However, such variation would not materially change the relative ranking of infectious diseases compared with other major health conditions. A further limitation relates to the use of GDP per worker as the basis for valuing productivity losses in the human capital approach. During the COVID-19 pandemic, exceptional fiscal measures—such as furlough schemes, direct transfers, and expanded public employment—partially decoupled GDP growth from actual labor input. As a result, GDP-based valuations may not fully reflect the true relationship between absenteeism and productivity losses in this period. Adjusting GDP for such distortions would require detailed fiscal and sectoral data that go beyond the scope of this study. Nevertheless, this issue should be acknowledged when interpreting pandemic-era cost estimates, as our results may overstate or understate the economic impact depending on how government interventions interacted with labor productivity.

Our study focuses on short-term productivity losses due to sickness absence, which provides only a partial view of the indirect costs associated with infectious diseases. We did not capture other relevant components, such as long-term disability, informal caregiving, presenteeism, or productivity impairment persisting after recovery. These omissions result from data limitations but should be acknowledged as important aspects of the broader economic and social burden of infectious diseases and should be the subject of future research.

Another limitation is that our analysis was restricted to individual-level absenteeism, without considering potential spillover effects on team or organizational productivity. In many work settings, the absence of one employee can reduce overall team output, redistribute workload, or increase stress for co-workers, thereby amplifying the productivity impact beyond the directly absent worker. Empirical evidence confirms that such “second-order” effects are particularly relevant for communicable diseases, where contagion risk can exacerbate disruptions at the workplace [44]. While not captured in our dataset, these effects represent an additional source of indirect costs that should be considered in future research.

Another important consideration is that our analysis of the 2018–2023 period was significantly impacted by the COVID-19 pandemic, whose consequences affected not only sickness absence but also the functioning of the entire healthcare system [45]. Non-pharmaceutical interventions (lockdowns, mask mandates, social distancing) temporarily reduced the circulation of many pathogens, potentially underestimating the typical burden of infectious diseases. At the same time, pandemic-related changes in healthcare utilization, diagnostic intensity, and reporting practices could have affected the consistency of sick leave records. In particular, we acknowledge the possibility of coding drift or misclassification in ICD-10 reporting during the early pandemic phase (e.g., frequent use of residual categories such as B34 “viral infection of unspecified site”). Although the Social Insurance Institution (ZUS) applies standardized coding procedures and provides training for certifying physicians, these factors cannot be fully excluded.

A further source of uncertainty concerns the COVID-19 period, when in some countries sick leave also covered caregiving or quarantine. In Poland, however, these categories were administered separately by ZUS and are not included in our dataset. Nevertheless, their coexistence with standard sick leave schemes may have contributed to some reporting heterogeneity, which should be borne in mind when interpreting pandemic-era results. Importantly, because our dataset spans 2018–2023, it covers both the pre-pandemic years and 2023, when restrictions were lifted in Poland and the pandemic’s impact on healthcare functioning was already minimal.

An important limitation is the omission of presenteeism, i.e., working despite being ill, which is particularly common in the health and social care sectors. Evidence shows that a considerable proportion of employees continue to work while experiencing infectious symptoms—systematic reviews report prevalence rates ranging from 35% to 97%, with higher levels observed among healthcare workers [40]. Similarly, Daniels et al. [41] found that the prevalence of work attendance despite confirmed or suspected respiratory infections varied widely across occupational settings. Presenteeism contributes both to hidden productivity losses and to further transmission of infectious diseases in the workplace. As such, its exclusion from our analysis likely results in an underestimation of the true economic and epidemiological burden. 

Ultimately, our findings reinforce the importance of renewed investment in prevention and workforce health resilience, particularly through evidence-based immunization programs. Prioritizing such interventions can deliver dual benefits: improving population health outcomes while safeguarding economic productivity. Responsibility in this area should not rest solely with the central government; local government units also play a complementary role, both in financing immunization programs and in promoting vaccination awareness [46].

## 5. Conclusions

This study analyzed six-year trends in productivity losses due to sickness absence from infectious diseases in Poland. Our findings show that infectious diseases are a major driver of work absence, with the impact strongly amplified by the COVID-19 pandemic. Moreover, the burden imposed by infectious diseases on the healthcare system and the wider economy is considerable, even when compared with other leading public health challenges such as malignant neoplasms, ischemic heart disease, and depression. These results underscore the need for sustained investment in effective public health interventions—particularly vaccination programs—to mitigate both the health and economic consequences of infectious diseases.

## Figures and Tables

**Figure 1 healthcare-13-02284-f001:**
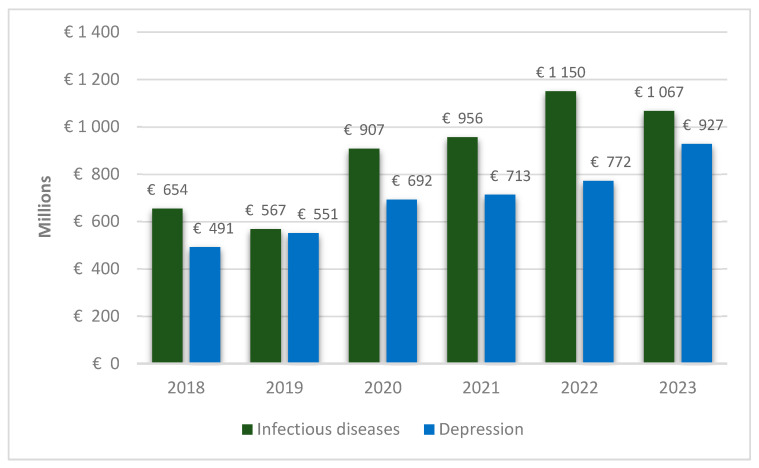
Comparison of sickness absence costs attributable to infectious diseases and depression in Poland (2018–2023).

**Figure 2 healthcare-13-02284-f002:**
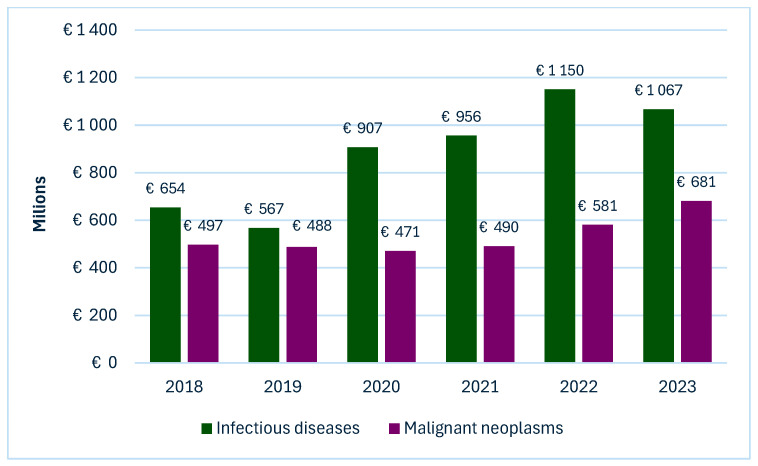
Comparison of sickness absence costs attributable to infectious diseases and malignant neoplasm in Poland (2018–2023).

**Figure 3 healthcare-13-02284-f003:**
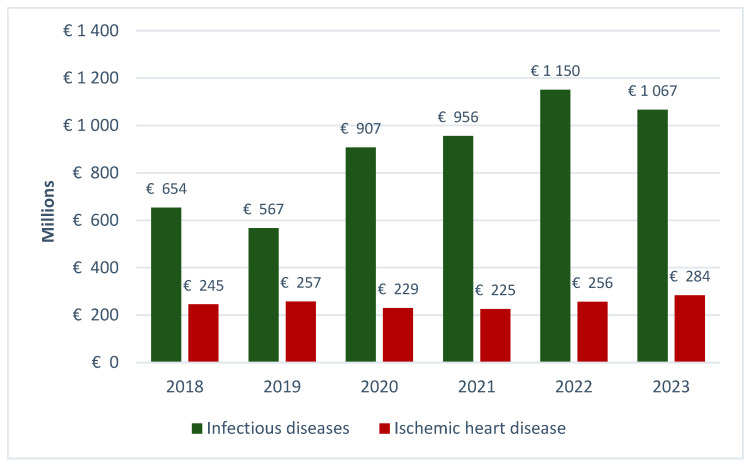
Comparison of sickness absence costs attributable to infectious diseases and ischemic heart disease in Poland (2018–2023).

**Table 1 healthcare-13-02284-t001:** Overview of socioeconomic parameters used in the model to estimate productivity losses due to infectious diseases in Poland from 2018 to 2023.

Year	Average EUR/PLN Exchange Rate	GDP (in EUR)	Employment	Assumed Number of Working Days per Year	GDP per Worker (Adjusted with CF)	Value of One Working Day
2018	4.2623	€ 497,496,656,735	16,409,000	227	€ 19,707	€ 86.81
2019	4.3018	€ 531,809,475,103	16,467,000	228	€ 20,992	€ 92.07
2020	4.4742	€ 520,016,092,262	16,555,000	230	€ 20,527	€ 89.25
2021	4.5652	€ 574,771,751,511	16,780,000	229	€ 22,265	€ 97.23
2022	4.6881	€ 654,315,180,990	16,796,000	228	€ 25,322	€ 111.06
2023	4.5430	€ 748,758,529,606	17,323,000	227	€ 28,095	€ 123.77

**Table 2 healthcare-13-02284-t002:** Sickness absences due to infectious diseases by ICD-10 code (2018–2020).

	2018		2019		2020	
ICD-10	No. of Sick Leave Days	No. of Issued Sick Leave Certificates	Number of Sick Leave Days	Number of Issued Sick Leave Certificates	Number of Sick Leave Days	Number of Issued Sick Leave Certificates
A00–A09: Intestinal infectious diseases (e.g., salmonella, rotavirus)	648,836	173,970	694,686	197,828	449,569	121,195
A15–A19: Tuberculosis	231,495	7856	183,922	6986	148,399	5756
A30-A49: Other bacterial diseases (e.g., whooping cough, diphtheria, tetanus)	163,011	11,750	143,287	11,382	109,984	8752
A50–A64: Infections with a predominantly sexual mode of transmission	11,847	1199	8815	995	6735	795
A80–A89: Viral infections of the central nervous system	46,756	2642	35,246	2464	20,576	1115
B00–B09: Viral infections with skin and mucous membrane lesions (e.g., varicella, herpes)	506,862	62,916	543,155	68,757	452,063	54,740
B15–B19: Viral hepatitis	150,788	12,337	99,410	9746	56,801	4742
B20–B24: HIV	26,796	1761	20,291	1565	17,354	1173
B25–B34: Other viral infections (e.g., cytomegalovirus, influenza-like illness, adenovirus)	80,848	14,223	93,480	17,740	757,523	111,861
B95–B97: Bacterial and viral agents as cause of diseases	5722	948	5364	990	33,062	4261
J09–J11: Influenza	1,222,976	184,225	681,099	117,729	669,537	105,428
U07–U09: COVID-19	n.a.	n.a.	n.a.	n.a.	4,835,175	617,906
J12–J22: Pneumonia and other acute lower respiratory tract infections	4,436,183	595,022	3,652,994	513,629	2,661,591	318,384
Total	7,532,120	1,068,849	6,161,749	949,811	10,218,369	1,356,108

n.a. means “not applicable”.

**Table 3 healthcare-13-02284-t003:** Sickness absences due to infectious diseases by ICD-10 code (2021–2023).

	2021		2022		2023	
ICD-10	Number of Sick Leave Days	Number of Issued Sick Leave Certificates	Number of Sick Leave Days	Number of Issued Sick Leave Certificates	Number of Sick Leave Days	Number of Issued Sick Leave Certificates
A00–A09: Intestinal infectious diseases (e.g., salmonella, rotavirus)	520,087	155,822	706,365	225,089	603,897	195,958
A15–A19: Tuberculosis	140,810	5467	159,786	6233	170,969	6466
A30–A49: Other bacterial diseases (e.g., whooping cough, diphtheria, tetanus)	92,582	7460	115,012	9112	146,911	12,022
A50–A64: Infections with a predominantly sexual mode of transmission	7732	978	8209	1071	7942	1234
A80–A89: Viral infections of the central nervous system	23,671	2063	41,770	4930	39,916	3257,000
B00–B09: Viral infections with skin and mucous membrane lesions (e.g., varicella, herpes)	358,186	47,340	427,682	58,493	432,638	60,504
B15–B19: Viral hepatitis	34,731	3808	38,723	4417	40,538	5235
B20–B24: HIV	16,724	1244	17,773	1511	20,275	1758
B25–B34: Other viral infections (e.g., cytomegalovirus, influenza-like illness, adenovirus)	431,428	87,368	536,578	121,900	518,635	126,662
B95–B97: Bacterial and viral agents as cause of diseases	27,092	3673	11,719	1950	12,498	2254
J09–J11: Influenza	141,995	24,232	622,188	115,588	1,077,152	198,951
U07–U09: COVID-19	5,404,746	652,002	4,923,944	737,221	2,560,970	407,646
J12–J22: Pneumonia and other acute lower respiratory tract infections	2,632,891	321,987	2,749,099	385,331	2,987,672	436,289
Total	9,832,675	1,313,444	10,358,848	1,672,846	8,620,013	1,458,236

**Table 4 healthcare-13-02284-t004:** Productivity losses associated with infectious disease-related sickness absence in Poland (2018–2023).

ICD-10	2018	2019	2020	2021	2022	2023	Total
A00–A09: Intestinal infectious diseases (e.g., salmonella, rotavirus)	€ 56,328,799	€ 63,960,027	€ 39,908,874	€ 50,565,852	€ 78,449,253	€ 74,742,746	€ 363,955,551
A15–A19: Tuberculosis	€ 20,097,275	€ 16,933,775	€ 13,173,588	€ 13,690,359	€ 17,745,914	€ 21,160,384	€ 102,801,294
A30–A49: Other bacterial diseases (e.g., whooping cough, diphtheria, tetanus)	€ 14,151,825	€ 13,192,493	€ 9,763,435	€ 9,001,355	€ 12,773,291	€ 18,182,789	€ 77,065,188
A50–A64: Infections with a predominantly sexual mode of transmission	€ 1,028,499	€ 811,601	€ 597,875	€ 751,750	€ 911,696	€ 982,960	€ 5,084,381
A80–A89: Viral infections of the central nervous system	€ 4,059,129	€ 3,245,114	€ 1,826,561	€ 2,301,431	€ 4,638,997	€ 4,940,299	€ 21,011,530
B00–B09: Viral infections with skin and mucous membrane lesions (e.g., varicella, herpes)	€ 44,003,304	€ 50,008,506	€ 40,130,270	€ 34,824,905	€ 47,498,579	€ 53,546,469	€ 270,012,032
B15–B19: Viral hepatitis	€ 13,090,684	€ 9,152,720	€ 5,042,305	€ 3,376,748	€ 4,300,596	€ 5,017,282	€ 39,980,334
B20–B24: HIV	€ 2,326,299	€ 1,868,201	€ 1,540,539	€ 1,626,004	€ 1,973,878	€ 2,509,384	€ 11,844,304
B25–B34: Other viral infections (e.g., cytomegalovirus, influenza-like illness, adenovirus)	€ 7,018,832	€ 8,606,742	€ 67,246,385	€ 41,945,914	€ 59,592,623	€ 64,190,092	€ 248,600,588
B95–B97: Bacterial and viral agents as cause of diseases	€ 496,756	€ 493,866	€ 2,934,960	€ 2,634,040	€ 1,301,518	€ 1,546,845	€ 9,407,985
J09–J11: Influenza	€ 106,172,852	€ 62,709,067	€ 59,435,744	€ 13,805,571	€ 69,100,513	€ 133,316,275	€ 444,540,022
U07–U09: COVID-19	n.a.	n.a.	€ 429,287,756	€ 525,480,521	€ 546,855,703	€ 316,964,532	€ 1,818,588,513
J12–J22: Pneumonia and other acute lower respiratory tract infections	€ 385,127,920	€ 336,332,669	€ 236,273,186	€ 255,984,821	€ 305,316,321	€ 369,776,318	€ 1,888,811,235
Total	€ 653,902,174	€ 567,314,780	€ 907,161,479	€ 955,989,271	€ 1,150,458,881	€ 1,066,876,374	€ 5,301,702,958

n.a. means “not applicable”.

**Table 5 healthcare-13-02284-t005:** Disability-Adjusted Life Years (DALYs) and their components (YLLs and YLDs) in Poland, 2021 (% of total), for communicable vs. non-communicable diseases (GBD 2021).

Measure	Communicable Diseases	Non-Communicable Diseases
DALYs (Disability-Adjusted Life Years) total	14.9%	74.4%
YLDs (Years Lived with Disability)	3.4%	85.0%
YLLs (Years of Life Lost)	13.8%	69.1%

## Data Availability

The data presented in this study are openly available in ZUS Database at https://www.zus.pl/baza-wiedzy/statystyka/opracowania-tematyczne/absencja-chorobowa, (accessed on 8 September 2025).

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
