# Peer review of "Which Infectious Diseases Drive the Highest Absenteeism Costs—An Analysis Based on National Data Covering the Entire Polish Population in the Period of 2018–2023"

_healthcare, 2025, doi:10.3390/healthcare13182284_

Round 1
Reviewer 1 Report
Comments and Suggestions for Authors
It is my pleasure to review the manuscript on “Which infectious diseases drive the highest absenteeism costs? An analysis based on national data covering the entire Polish population in the period of 2018–2023”.
In my opinion, the manuscript provides valuable insights into the economic burden of infectious diseases in Poland and highlights the importance of preventive efforts in mitigating this burden.
The manuscript articulates the research question and objective, which is to quantify the indirect costs of infectious disease-related sick leave in Poland. It employs a societal perspective to capture productivity losses reflected in reductions in gross domestic product (GDP), a comprehensive approach to estimating the economic burden of infectious diseases.
Further, the study highlights the importance of investing in effective public health interventions, particularly vaccination programs, to mitigate both the health and economic impacts of infectious diseases.
However, the following issues and clarifications should be considered for the improvement of this manuscript : -
- Lines 44-49 need more description with adequate references from a review of studies
- Lines 53-55 – These lines are so broad that they need to be supported with justifications. The authors should provide the meaning of the full economic burden. Does absenteeism only cause economic burden? What about other indirect costs?
- Lines 57-58 – the authors should clarify what social and economic costs are associated with infectious diseases
- The introduction should be supported with an adequate review of existing literature and its gaps.
- The research uses the human capital approach, which may overestimate productivity losses compared to other methods, such as the friction cost method. The authors should clarify the reasons for choosing human capital approach compared to other methods.
- Authors compare the costs of infectious diseases with those associated with other major public health challenges, like cancer and ischemic heart disease. It would be useful to provide more context on the methodology used for these comparisons.
- The major focus is on short-term absenteeism, providing only a partial view of the indirect costs associated with infectious diseases, and ignoring long-term disability, informal caregiving, and productivity impairment post-recovery etc. Authors should clarify these issues. The shame should also be highlighted as a limitation.
- The authors should provide a critical analysis of the impact of the COVID-19 pandemic on the results, including potential biases and uncertainties, especially using this data.
Author Response
- Lines 44-49 need more description with adequate references from a review of studies
Response: Thank you for your valuable input. We expanded this section to provide a more detailed description of the direct and indirect costs associated with infectious diseases, and we supported the text with additional references from recent systematic reviews. These changes can be found in the revised manuscript at Lines 49-58
- Lines 53-55 – These lines are so broad that they need to be supported with justifications. The authors should provide the meaning of the full economic burden. Does absenteeism only cause economic burden? What about other indirect costs?
Response: We appreciate this helpful suggestion. In response, we clarified our definition of the “full economic burden” to explicitly include direct medical costs, direct non-medical costs, indirect costs (absenteeism, presenteeism, premature mortality, and disability), and intangible costs. We also expanded the text with concrete examples and additional literature, including Polish evidence. These revisions are included in Lines 66–77 of the manuscript.
- Lines 57-58 – the authors should clarify what social and economic costs are associated with infectious diseases
The introduction should be supported with an adequate review of existing literature and its gaps
Response: Thank you for your constructive comment. We revised this section to clarify what we mean by “social and economic costs” of infectious diseases, highlighting costs to the healthcare system, households, employers, and broader macroeconomic impacts. We also added recent European and national evidence to strengthen the argument. The revised text is now presented in Lines 81-93
- The research uses the human capital approach, which may overestimate productivity losses compared to other methods, such as the friction cost method. The authors should clarify the reasons for choosing human capital approach compared to other methods.
Response: We thank the reviewer for this important suggestion. In the Methodology section (Lines 125-134) we clarified that the human capital approach (HCA) was chosen because it is explicitly recommended by the Polish HTA guidelines and is consistent with the structure of national data sources (ZUS, GUS). In addition, we have addressed the reviewer’s concern about potential overestimation by HCA compared with the friction cost method (FCM) in the Discussion (Lines 416–423), where we explain that, although HCA may yield higher estimates, in our study this risk was mitigated by conservative assumptions (exclusion of presenteeism, informal caregiving, and intangible costs).
- Authors compare the costs of infectious diseases with those associated with other major public health challenges, like cancer and ischemic heart disease. It would be useful to provide more context on the methodology used for these comparisons.
Response: Thank you for this helpful comment. We revised the Methodology section (Lines 171–181) to clarify the basis for these comparisons, including the criteria for selecting the reference conditions and the consistency of data sources and analytical approach across all disease categories.
- The major focus is on short-term absenteeism, providing only a partial view of the indirect costs associated with infectious diseases, and ignoring long-term disability, informal caregiving, and productivity impairment post-recovery etc. Authors should clarify these issues. The shame should also be highlighted as a limitation.
Response: We thank the reviewer for this comment. We have expanded the Discussion (Lines 444–450) to explicitly acknowledge that our analysis captures only short-term absenteeism and does not account for long-term disability, informal caregiving, presenteeism, or post-recovery productivity impairment. We now highlight this as a limitation of the study, which results from the scope of available data.
- The authors should provide a critical analysis of the impact of the COVID-19 pandemic on the results, including potential biases and uncertainties, especially using this data.
Response: We appreciate this important suggestion. In the Discussion (Lines 460–471), we now provide a critical reflection on how the COVID-19 pandemic may have affected both the incidence of infectious diseases and sickness absence reporting, introducing potential sources of bias. We also emphasize that our dataset spans 2018–2023, thereby covering both the pre-pandemic years and 2023, when restrictions were no longer in place and the pandemic’s impact on healthcare functioning in Poland was minimal. This broader perspective helps mitigate the risk of pandemic-related bias dominating our findings.
Reviewer 2 Report
Comments and Suggestions for Authors
- Summary
Based on information on sickness leaves, the manuscript aims to determine which infectious diseases caused the highest costs associated with absenteeism from work. The case study is Poland, between 2018 and 2023. Pneumonia and other acute lower respiratory tract infections, CoViD-19, and influenza were identified as the infectious diseases that caused the highest costs of that type. These costs were higher than those associated with malignant neoplasms, depression, and ischemic heart disease.
- Appraisal
I believe that the topic discussed in the manuscript is relevant and appropriate for the Journal. This topic is addressed using a correct methodology, which was applied without errors. The manuscript’s structure is the one typically used in scientific articles. Moreover, it is complete, and easy to be read.
My first recommendation concerns the magnitude of the global burden of disease figures. I acknowledge that the manuscript intends to study the costs of work absenteeism, and these are the ones that led to the conclusions, which are correct. Even so, I believe it would be interesting to contextualize these costs (economic and, essentially, short- and medium-term) in relation to the costs (of various categories and over various terms) measured by DALYs and their components, YLDs and YLLs. To be clearer, consider Table 1 – please see the pdf of this report –, in which these figures are presented. The authors might want to detail the diseases.
Table 1. Global Burden of Disease in Poland (all ages & both sexes)
measure_name |
cause_name |
metric_name |
year |
val |
upper |
lower |
DALYs (Disability-Adjusted Life Years) |
Communicable, maternal, neonatal, and nutritional diseases |
Percent |
2021 |
0,14876 |
0,18773 |
0,11602 |
DALYs (Disability-Adjusted Life Years) |
Non-communicable diseases |
Percent |
2021 |
0,74365 |
0,77921 |
0,71183 |
YLDs (Years Lived with Disability) |
Communicable, maternal, neonatal, and nutritional diseases |
Percent |
2021 |
0,05342 |
0,07904 |
0,04092 |
YLDs (Years Lived with Disability) |
Non-communicable diseases |
Percent |
2021 |
0,84589 |
0,86795 |
0,81637 |
YLLs (Years of Life Lost) |
Communicable, maternal, neonatal, and nutritional diseases |
Percent |
2021 |
0,19799 |
0,25418 |
0,15234 |
YLLs (Years of Life Lost) |
Non-communicable diseases |
Percent |
2021 |
0,69088 |
0,73780 |
0,64282 |
DALYs (Disability-Adjusted Life Years) |
Communicable, maternal, neonatal, and nutritional diseases |
Rate |
2021 |
6027,28405 |
7428,77467 |
4745,56354 |
DALYs (Disability-Adjusted Life Years) |
Non-communicable diseases |
Rate |
2021 |
30199,59597 |
33879,51673 |
26733,45909 |
YLDs (Years Lived with Disability) |
Communicable, maternal, neonatal, and nutritional diseases |
Rate |
2021 |
741,73508 |
1127,02297 |
494,32531 |
YLDs (Years Lived with Disability) |
Non-communicable diseases |
Rate |
2021 |
11753,53520 |
15303,15503 |
8765,88935 |
YLLs (Years of Life Lost) |
Communicable, maternal, neonatal, and nutritional diseases |
Rate |
2021 |
5285,54897 |
6764,92168 |
4075,76167 |
YLLs (Years of Life Lost) |
Non-communicable diseases |
Rate |
2021 |
18446,06077 |
19717,30053 |
17176,93779 |
Source: Global Burden of Disease Study 2021 (GBD 2021) Results. Seattle, United States: Institute for Health Metrics and Evaluation (IHME), 2022. Available from https://vizhub.healthdata.org/gbd-results/
My second recommendation concerns the robustness of the results. The authors considered a labor productivity correction factor of 0.65, and this choice was well-founded. Even so, it would have been interesting to examine the sensitivity of the results to other values (around that) for that factor.
Finally, please note the typo in the caption of Figure 1: “[…] (2018–202)” and some incongruences in referencing format, in the list of references.
Author Response
. My first recommendation concerns the magnitude of the global burden of disease figures. I acknowledge that the manuscript intends to study the costs of work absenteeism, and these are the ones that led to the conclusions, which are correct. Even so, I believe it would be interesting to contextualize these costs (economic and, essentially, short- and medium-term) in relation to the costs (of various categories and over various terms) measured by DALYs and their components, YLDs and YLLs. To be clearer, consider Table 1 – please see the pdf of this report –, in which these figures are presented. The authors might want to detail the diseases.
Response: Thank you for this valuable comment. In response, we have clarified in the Methodology (lines 182–185) how the Global Burden of Disease (GBD) data were retrieved and applied for contextual comparison. Furthermore, we incorporated a new subsection titled “Infectious Diseases as a Public Health Challenge: Comparison with Other Major Conditions” (lines 286–302), where we present and discuss GBD 2021 estimates (DALYs, YLLs, YLDs) for Poland. This addition enables us to contextualize the indirect costs of absenteeism against the broader population health burden of communicable and non-communicable diseases.
- My second recommendation concerns the robustness of the results. The authors considered a labor productivity correction factor of 0.65, and this choice was well-founded. Even so, it would have been interesting to examine the sensitivity of the results to other values (around that) for that factor.
Response: Thank you for this insightful comment. We agree that exploring alternative values for the labor productivity correction factor could provide additional robustness. In our revised manuscript, we explicitly acknowledge this issue in the Discussion (lines 429–433), noting that while we applied the widely used value of 0.65—consistent with Polish and European cost-of-illness studies—different assumptions (e.g. 0.6–0.7) might shift absolute estimates but would not alter the relative ranking of infectious diseases compared with other major conditions.
- Finally, please note the typo in the caption of Figure 1: “[…] (2018–202)” and some incongruences in referencing format, in the list of references.
Response: We thank the Reviewer for carefully checking the manuscript. The typo in the caption of Figure 1 (“2018–202”) has been corrected to “2018–2023.” We also revised the reference list to ensure consistency in formatting.
Reviewer 3 Report
Comments and Suggestions for Authors
The manuscript presents a timely and data-rich analysis of the indirect economic burden of infectious disease-related absenteeism in Poland, using national-level administrative data and the human capital approach. The clarity of methodology and contextual comparison with chronic diseases are commendable. However, several conceptual and methodological concerns warrant clarification or revision to strengthen the interpretation and utility of the findings.
- The authors should clarify whether all sickness absences classified under infectious diseases (ICD-10 codes) reflect personal illness alone, or whether they also include quarantine mandates, caregiving duties (e.g. for ill minors), or preventive absences. Especially during the COVID-19 pandemic, some countries issued sick leave for non-ill caregivers or for isolation after exposure. If the ZUS database includes such categories, the absenteeism burden may not purely reflect disease incidence or severity. Please detail how such cases were coded and whether the database differentiated between direct illness and quarantine/caregiving-related leave. A sensitivity analysis excluding such categories, if identifiable, would be beneficial.
- The use of GDP per worker in the human capital approach should be further contextualized, given the pandemic's dual impact on employment and government expenditure. The pandemic period saw exceptional public expenditure (e.g., stimulus packages, furlough schemes) that partially decoupled productivity from labor input. This could affect GDP-based valuations. The authors might discuss whether adjusting GDP for pandemic-related distortions (e.g., subtracting COVID-related fiscal transfers or government employment effects) would yield more accurate absenteeism costs. Alternatively, briefly acknowledging these limitations would improve transparency.
- Epidemiological reasons may justify the choice of comparators (depression, ischaemic heart disease, and malignant neoplasms), but their comparability in terms of sickness absence costs is uneven. Depression is known to be underdiagnosed and stigmatised (Sheehan, 2004; Bertilsson et al., 2022), leading to underreporting and lower sick leave registration. Malignant neoplasms often result in permanent disability or expulsion from the labour market, not short-term absenteeism (Horsbøl, 2014). IHD episodes may result in acute leave but also long-term disability or presenteeism (Saarinen et al., 2024). Please discuss the limitations of using these comparators strictly on the basis of short-term sickness absence. Your framework likely underestimates indirect costs from presenteeism, early retirement, or informal care for these conditions.
- The study focuses solely on individual-level absenteeism, but infectious diseases often have spillover effects on team or organisational productivity. In many work settings, the absence of one employee can reduce team output or increase workload and stress for others. These "second-order" losses are relevant for communicable diseases (Koopmanschap et al., 2013). Consider acknowledging this limitation in the Discussion section. Reference to relevant literature on team productivity and contagion effects would enrich the interpretation.
- There may be misclassification or inconsistent use of ICD-10 codes, especially during the early pandemic period (e.g., use of “B34—“Viral infection of unspecified site”). It would be useful to explain how coding practices were standardized over the years and whether training or audit procedures were in place for physicians reporting to ZUS. A brief mention of potential coding drift or overuse of residual categories could improve methodological transparency.
- Additional Suggestions:
- Use consistent terminology for sick leave (e.g., “sickness absence” vs. “work absence”) throughout for clarity.
- The limitations section might benefit from a stronger mention of presenteeism especially in health/social sectors, where infectious disease-related work attendance remains common.
With clearer methodological delineation and deeper contextual discussion of the limitations around coding, comparator diseases, and GDP adjustments, it would offer more robust and actionable findings. For now, my decision is a MAJOR REVISION. Please also consider providing minimal data examples (actual data or code) so reviewers can test boyong that is provided in the article.
BR
Author Response
- The authors should clarify whether all sickness absences classified under infectious diseases (ICD-10 codes) reflect personal illness alone, or whether they also include quarantine mandates, caregiving duties (e.g. for ill minors), or preventive absences. Especially during the COVID-19 pandemic, some countries issued sick leave for non-ill caregivers or for isolation after exposure. If the ZUS database includes such categories, the absenteeism burden may not purely reflect disease incidence or severity. Please detail how such cases were coded and whether the database differentiated between direct illness and quarantine/caregiving-related leave. A sensitivity analysis excluding such categories, if identifiable, would be beneficial.
Response: Thank you for raising this important point. We confirm that the ZUS database used in our study captures only sickness absences certified by physicians with an ICD-10 code, reflecting the personal illness of the insured worker. According to our knowledge, ZUS records absences related to caregiving duties and quarantine/isolation mandates during the COVID-19 pandemic under separate schemes, and therefore these were not included in our dataset. We have clarified this in the Methodology (lines 135–140) and added a note in the Discussion (lines 472–479) to highlight the potential implications for interpreting pandemic-era results.
- The use of GDP per worker in the human capital approach should be further contextualized, given the pandemic's dual impact on employment and government expenditure. The pandemic period saw exceptional public expenditure (e.g., stimulus packages, furlough schemes) that partially decoupled productivity from labor input. This could affect GDP-based valuations. The authors might discuss whether adjusting GDP for pandemic-related distortions (e.g., subtracting COVID-related fiscal transfers or government employment effects) would yield more accurate absenteeism costs. Alternatively, briefly acknowledging these limitations would improve transparency.
Response: We fully agree with the Reviewer’s observation. In the revised Discussion section we have explicitly acknowledged this limitation and emphasized that GDP per worker may have been partially decoupled from actual labor input during the COVID-19 pandemic due to fiscal interventions. We also note that while adjusting GDP for such distortions would require detailed fiscal and sectoral data beyond the scope of this study, our estimates should be interpreted with this caveat in mind (lines 434–443).
- Epidemiological reasons may justify the choice of comparators (depression, ischaemic heart disease, and malignant neoplasms), but their comparability in terms of sickness absence costs is uneven. Depression is known to be underdiagnosed and stigmatised (Sheehan, 2004; Bertilsson et al., 2022), leading to underreporting and lower sick leave registration. Malignant neoplasms often result in permanent disability or expulsion from the labour market, not short-term absenteeism (Horsbøl, 2014). IHD episodes may result in acute leave but also long-term disability or presenteeism (Saarinen et al., 2024). Please discuss the limitations of using these comparators strictly on the basis of short-term sickness absence. Your framework likely underestimates indirect costs from presenteeism, early retirement, or informal care for these conditions.
Response: Thank you for your valuable comment highlighting the limitations of using depression, ischemic heart disease, and malignant neoplasms as comparators. We agree that comparability across these conditions is imperfect, as depression is prone to underdiagnosis and underreporting in sick leave records, malignant neoplasms often lead to early retirement or permanent exit from the labor market, and ischemic heart disease involves both acute and long-term productivity impairments. To address this point, we have added a new paragraph at the end of the section Infectious Diseases as a Public Health Challenge: Comparison with Other Major Conditions (lines 358–369), explicitly acknowledging these limitations and clarifying that comparisons restricted to short-term absenteeism are likely to underestimate the full indirect costs associated with these conditions.
In addition, to further contextualize our findings, we incorporated Global Burden of Disease (GBD 2021) estimates, which provide a broader epidemiological perspective and illustrate the relative burden of communicable versus non-communicable diseases beyond sickness absence alone (lines 286–302).
- The study focuses solely on individual-level absenteeism, but infectious diseases often have spillover effects on team or organisational productivity. In many work settings, the absence of one employee can reduce team output or increase workload and stress for others. These "second-order" losses are relevant for communicable diseases (Koopmanschap et al., 2013). Consider acknowledging this limitation in the Discussion section. Reference to relevant literature on team productivity and contagion effects would enrich the interpretation.
Response: Thank you for this insightful suggestion. In the Discussion (lines 451–459) we now acknowledge that focusing on individual absenteeism omits important spillover effects at the team and organizational level. As shown by Zhang et al. (2017), absenteeism among team-based workers can generate productivity losses beyond the individual level, reflecting the broader disruption of team output.
- There may be misclassification or inconsistent use of ICD-10 codes, especially during the early pandemic period (e.g., use of “B34—“Viral infection of unspecified site”). It would be useful to explain how coding practices were standardized over the years and whether training or audit procedures were in place for physicians reporting to ZUS. A brief mention of potential coding drift or overuse of residual categories could improve methodological transparency.
Response: Thank you for raising this important point. We expanded the Discussion (lines 460–471) to describe the broader impact of the COVID-19 pandemic on our analysis. This includes potential effects on healthcare utilization, diagnostic intensity, reporting practices, and, among them, the possibility of ICD-10 coding drift or misclassification in the early pandemic phase. By explicitly acknowledging these factors, we aimed to increase the transparency of our cost estimates and highlight pandemic-specific sources of uncertainty.
- Use consistent terminology for sick leave (e.g., “sickness absence” vs. “work absence”) throughout for clarity.
Response: We thank the Reviewer for this helpful comment. In line with the suggestion, we have standardized terminology throughout the manuscript and consistently use “sickness absence” as the preferred term.
- The limitations section might benefit from a stronger mention of presenteeism especially in health/social sectors, where infectious disease-related work attendance remains common.
Response: We thank the Reviewer for this valuable suggestion. We have revised the Limitations section to explicitly address the issue of presenteeism, particularly in health and social care sectors where working despite infection is prevalent. We emphasize that this behavior contributes to hidden productivity losses and facilitates further transmission, and its exclusion likely results in an underestimation of the true economic burden of infectious diseases (lines 480–489).
- Please also consider providing minimal data examples (actual data or code) so reviewers can test boyong that is provided in the article.
Response: We appreciate the Reviewer’s suggestion regarding the provision of minimal data examples. We would like to emphasize that all readers can easily reproduce the core cost estimates using the data already provided in the manuscript. Specifically, the value of one working day (Table 1, Methodology) multiplied by the number of sickness absence days (Tables 2 and 3, Sickness Absences Due to Infectious Diseases by ICD-10 Code) directly yields the social cost estimates. As presented in Table 4 (Productivity losses associated with infectious disease-related sickness absence in Poland, 2018–2023), our calculations were performed using the value of a working day specified to several decimal places, whereas in Table 1 we reported this value rounded to two decimal places. Consequently, minor discrepancies may appear when reproducing the results, but these do not affect the substantive conclusions of the study.
Round 2
Reviewer 3 Report
Comments and Suggestions for Authors
The reference number and US/UK spelling shall go under minor edits. Otherwise, after that revision, I have no further comments or suggestions. Good luck.
Author Response
The reference number and US/UK spelling shall go under minor edits.
We would like to confirm that both minor revisions requested by the reviewer have been addressed:
- The reference numbering has been checked and corrected to ensure full consistency between the text and the reference list.
- The manuscript has been revised to use consistent US English spelling throughout.